# A Systematic Review of the Clinical Use of Gabapentin and Pregabalin in Bipolar Disorder

**DOI:** 10.3390/ph14090834

**Published:** 2021-08-24

**Authors:** Qin Xiang Ng, Ming Xuan Han, Seth En Teoh, Clyve Yu Leon Yaow, Yu Liang Lim, Kuan Tsee Chee

**Affiliations:** 1MOH Holdings Pte Ltd., 1 Maritime Square, Singapore 099253, Singapore; yulianglim95@gmail.com; 2Department of Community Emergency Health and Paramedic Practice, Monash University, Clayton, VIC 3800, Australia; mxhan9598@yahoo.com; 3Yong Loo Lin School of Medicine, National University of Singapore, 10 Medical Dr, Singapore 117597, Singapore; e0659260@u.nus.edu (S.E.T.); e0268630@u.nus.edu (C.Y.L.Y.); 4Department of General and Community Psychiatry, Institute of Mental Health, 10 Buangkok View, Buangkok Green, Medical Park, Singapore 539747, Singapore; kuan_tsee_chee@imh.com.sg

**Keywords:** bipolar disorder, psychopharmacology, gabapentinoids, gabapentin, pregabalin

## Abstract

Despite its prevalence and disease burden, several chasms still exist with regard to the pharmacotherapy of bipolar disorder (BD). Polypharmacy is commonly encountered as a significant proportion of patients remain symptomatic, and the management of the depressive phase of the illness is a particular challenge. Gabapentin and pregabalin have often been prescribed off-label in spite of a paucity of evidence and clinical practice guidelines to support its use. This systematic review aimed to synthesize the available human clinical trials and inform evidence-based pharmacological approaches to BD management. A total of six randomized, controlled trials (RCTs) and 13 open-label trials involving the use of gabapentin and pregabalin in BD patients were reviewed. Overall, the studies show that gabapentin and its related drug pregabalin do not have significant clinical efficacy as either monotherapy or adjunctive therapy for BD. Gabapentin and pregabalin are probably ineffective for acute mania based on the findings of RCT, with only small open-label trials to support its potential adjunctive role. However, its effects on the long-term outcomes of BD remain to be elucidated. The evidence base was significantly limited by the generally small sample sizes and the trials also had heterogeneous designs and generally high risk of bias.

## 1. Introduction

Bipolar disorder (BD) is a debilitating mental illness that affects more than 1% of the world’s population [1]. Its lifetime prevalence in adults across 11 countries was estimated to be 0.4% [2]. In adolescents, the prevalence rate increases to 3–4% [2], making it one of the main causes of disability among youth [1]. In most patients, the onset of cognitive and psychosocial decline begins often at an age younger than 30 years [3] and is characterized by symptoms of depression and mania (bipolar I) or depression and hypomania (bipolar II) [1,3]. This predisposes the individual with BD to a significantly higher risk of death by suicide [4], an unfortunate clinical outcome that remains a challenging and pertinent issue [5]. It has also been suggested that sensory processes unique to individuals are implicated in their corresponding emotional patterns, making BD a very heterogenous condition [6].

The heavy socioeconomic burden associated with BD cannot be underestimated. Costs per capita ranged from USD 4000 to 5000 for direct mental healthcare and from USD 8000 to 14,000 for overall direct healthcare [7]. In the United States alone, the total costs of bipolar I disorder were approximately USD 200 billion in 2015 [8].

Therapeutic measures for BD are unfortunately limited by the incomplete remission of symptoms and frequent relapses. The Systematic Treatment Enhancement Program for Bipolar Disorder (STEP-BD) study reported recurrence rates of more than 50% and recovery rates of lower than 60% [9]. There is a glaring paucity of treatment options for bipolar depression and a clear need for effective acute and maintenance treatments for all individuals with BD, in order to delay illness progression, restore functioning and improve quality of life [10].

In modern clinical practice, patients generally are started on first-line BD medications and depending on symptom improvement and tolerability, either continued on the treatment regimen (with appropriate dose titration) or progressed to second-line medications. Alas, a combination of antipsychotics, mood stabilizers and other classes of psychotropic medications is often the choice in this challenging patient population, despite the inconsistent and scant evidence for polypharmacy [11].

With regard to pharmacotherapy, there has been sustained interest in the (off-label) use of gabapentin and its active metabolite, pregabalin in BD management. In patients with BD, the calcium pathway in intracellular secondary messaging of platelets is heightened [12], which suggests the involvement of calcium channels in BD pathophysiology. Pharmacodynamic evidence classifies gabapentin and pregabalin as ligands of the alpha-2-delta subunit of voltage-gated calcium channels [13], and they have been reported to decrease neocortical noradrenaline release [14]. This inhibitory effect on calcium currents attenuates neurotransmitter release and subsequently reduces postsynaptic excitation [15]. While this mechanism of action has been reported in both rodent and human models, and across different pathological states (epilepsy, pain, anxiety) [14], evidence of efficacy for gabapentinoids in BD is still limited [16].

This systematic review hence endeavors to synthesize and elucidate all available evidence of gabapentin and pregabalin in the treatment of BD, thereby informing evidence-based pharmacological approaches to BD management.

## 2. Methods

This review protocol was guided by the latest Preferred Reporting Items for Systematic Reviews and Meta-Analyses (PRISMA) guidelines [17]. A systematic search strategy employing different combinations of the keywords (bipolar, mania, hypomania, gabapentin, neurontin, gralise, gabarone, fanatrex, pregabalin, lyrica) was developed and performed in five databases namely OVID Medline, PubMed, ProQuest, PsychInfo and ScienceDirect from database inception to 7 June 2021. A search of gray literature was also employed to maximize identification of articles of interest. Abstracts were imported into Covidence (Melbourne, Victoria, Australia) and screened by three independent researchers (Q.X.N., M.X.H., Y.L.L). Full-text articles were obtained for all abstracts of relevance and their respective reference lists hand-searched for references of interest. Forward searching of prospective citations of the relevant full texts was also performed and authors of the articles were contacted if necessary to provide additional data.

Full-text articles which were obtained for all relevant abstracts were reviewed by three researchers (Q.X.N., M.X.H., Y.L.L) for inclusion. Any disagreement was resolved by discussion and consensus. Studies were deemed eligible for inclusion based on the following criteria: (i) original published prospective clinical trial, (ii) patients were clinically diagnosed with bipolar disorder. Studies which were unpublished and not in English were excluded along with all case series, case reports, reviews, opinions and comments.

Data such as study design and population, clinical assessment tools, pharmacological interventions and key findings were extracted from all the studies reviewed and are summarized in Table 1. The quality and risk of bias of studies was assessed using the Cochrane Collaboration’s tool for assessing risk of bias in randomized trials [18], and graded based on the consensus of three study investigators (Q.X.N., M.X.H., Y.L.L.).

## 3. Results

Figure 1 detailed the study selection and identification process. A total of 1186 records were found from the database search, with 767 records marked ineligible by automated filters and 114 records removed as duplicates. A total of 286 articles were further excluded after title and abstract screening, and subsequently three articles were excluded after a full text review. Finally, a total of 19 studies were included for thematic analysis [19,20,21,22,23,24,25,26,27,28,29,30,31,32,33,34,35,36,37].

There were 18 studies that trialled gabapentin use in BD [16,17,18,19,20,21,22,23,24,25,26,27,28,30,31,32,33,34] and only one study on pregabalin [29]. With reference to Table 1, study designs were heterogenous with a total of 13 open-label trials [19,21,22,23,25,26,30,31,32,33,34,36,37] and six RCTs, which employed different treatment regimes from cross-over trials to fixed-dose trials. [20,24,27,28,29,35] Most of these trials had a generally high risk of bias, as seen in Table 2. Due to this heterogeneity in designs, it was challenging to discern the source of the therapeutic effect, making it difficult to attribute any observed benefit to solely gabapentin or pregabalin. A meta-analysis was not performed for these reasons.

### 3.1. Intervention Type

Most of the studies were adjunctive treatment trials: 12 of the included studies administered gabapentin as an adjunctive treatment [16,17,18,19,23,27,28,30,31,32,33,34]. One study administered pregabalin as an adjunctive treatment [29]. Three studies administered gabapentin as a monotherapy [20,22,26] and another two studies employed a cross-over design with lamotrigine and placebo with a one-week washout period in between treatment segments [21,25]. Only one study administered gabapentin both as an add-on treatment and a monotherapy and compared the differences between these two experimental arms [20].

### 3.2. Dosing Regimes

The most common range of gabapentin dosing was 300 to 2400 mg/day. The maximum daily dose was 4800 mg. It was reported by Sokolski et al. that most patients had attained therapeutic effectiveness at a 900 mg dose even though the initial dose was 300 mg [33]. For the sole study on pregabalin, the average dose for acute patients was 72 mg/day [29].

### 3.3. Clinical Assessment Tools

There was a wide spectrum of scales used to assess symptom severity and treatment response across all 19 studies. With respect to the assessment of mania, ten studies utilized the YMRS, three studies utilized the BMRS while another three studies used the HAM-A. The Clinical Global Impression Scale for use in Bipolar Illness (CGI-BP) was used in five studies while the CGI-S and CGI-C subscales were used in two and three studies respectively. The BPRS was used in three studies, while the STAI was used in two studies. The AIRP, MMPI-2, SADS and ISS were each employed by one study.

### 3.4. Treatment Efficacy and Side Effects

Five studies reported a significant reduction in severity scores for mania post gabapentin therapy [20,23,27,29,33]. Among these five trials, it was Pande et al.’s 2000 study that reported a lower decrease in YMRS scores for the gabapentin experimental arm as compared to the placebo arm [29].

Seven studies reported a significant reduction in severity scores for depression, with gabapentin therapy. Four studies reported significant improvement in bipolar severity as measured by BPRS, AIRP and CGI-BP. Sedation was the most common side effect as reported in six studies [19,21,23,26,33,36].

Of note is the difference in treatment efficacy noted by Erfurth et al.’s 1998 study that compared adjunctive gabapentin treatment with gabapentin monotherapy within the same time period [20]. For the adjunctive group, all six patients had a significant decrease in their BMRS scores, while for the monotherapy group, four out of eight patients dropped out due to treatment insufficiency. There was a significant decrease in BMRS scores for the remaining four patients in the monotherapy group.

## 4. Discussion

Overall, the studies show that gabapentin and its related drug pregabalin do not have significant clinical efficacy as either monotherapy or adjunctive therapy for patients with BD. Multiple RCTs have found that gabapentin and pregabalin are ineffective for acute mania, with only small open-label trials to support its potential adjunctive role. There may be an adjunctive role for patients in a depressed state or with comorbid anxiety or substance use issues.

As most studies were adjunctive treatment trials, it remains unclear if the positive therapeutic response observed was primarily the result of the drug that was added to pre-existing agents (usually stable doses of mood stabilizers or antipsychotics), or the result of a synergistic effect. Furthermore, in open trials, without appropriate controls, it is also unclear if the observed effect is due to spontaneous remission of symptoms given the natural history of BD. While crossover trial designs have a key drawback of carryover effects although these tend to be reduced with a washout period that lasts a week or two between treatment segments [38].

The existence of a single trial that investigated the use of pregabalin in BD limits any strong conclusions. Pregabalin was developed as a successor to gabapentin; it was formally approved in 2004, as compared to gabapentin, which has been in use since 1994 [39]. The pharmacodynamic action of pregabalin is similar to gabapentin, characterized by its binding to the α2-δ subunit on voltage-gated calcium channels. Pregabalin is structurally similar to the inhibitory neurotransmitter γ-aminobutyric acid (GABA) although it does not bind to GABA receptors nor does it impact the uptake or breakdown of GABA. While various pharmacologic effects have been reported, pregabalin essentially acts to reduce the release of excitatory neurotransmitters such as substance P and glutamate, which have been linked to the pathogenesis of bipolar disorders.

Gabapentin and pregabalin are probably not effective for depressive state but may improve some subscales, such as irritability, social withdrawal or anxiety. They may have benefits for anxiety symptoms as do most GABAergic agents and also potential utility for bipolar individuals with comorbid substance use disorders. Its sedating effects probably can help alleviate insomnia and it is generally well-tolerated in terms of side-effect profile.

Bipolar illness is a life-course disorder; its chronic, enduring nature with interspersed periods of elation, irritability and depression usually demands maintenance treatment [10]. It was interesting to note that in an open-label trial, 39.5% (*n* = 17) of the patients maintained symptomatic remission over a period of 4 to 18 months on adjunctive gabapentin [28]. However, as therapeutic strategies shift towards long-term horizons, there are increasing concerns regarding dependence with long-term use. Relative to lithium, the use of gabapentin is significantly associated with a doubling of the risk of suicidality in patients diagnosed with BD [40].

Potential issues with dependence and also elevated mood switch belie the use of gabapentinoids. Even though gabapentin and pregabalin are considered as a treatment option for alcohol and substance abuse, there are numerous published case reports and case series documenting abuse, dependence, and withdrawal effects [41]. Caution must also be exercised by monitoring renal function due to the excretion of these drugs via renal pathways.

Another limitation of current evidence pertains to the different clinical assessment tools used to evaluate BD symptom severity and treatment response. BD is in itself a complex disorder, presenting with heterogeneous symptoms ranging from depression, hypomania to mixed states and even psychosis [42]. Many rating tools have been used in the clinical assessment of BD patients; however, they all have certain weaknesses [43]. For example, the usual CGI is a global measure of improvement in functioning, without rating scales specific for hypomanic/manic and depressive symptoms, compared to the Young Mania Rating Scale (YMRS), the Bech–Rafaelsen Mania Rating Scale (BMRS) [44], or the modified CGI-BP [45]. When studying patients with rapid-cycling states, multiple scales should be used to more adequately evaluate response [24]. Importantly, the contemporary redefinition of the clinical hallmarks of bipolar disorder (with activation as the most common dimension in mania), also necessitates the revisiting of new scales that apportion greater emphasis to activity or energy levels in this patient population [46,47].

## 5. Conclusions

In conclusion, there is a lack of rigorous evidence to support the clinical efficacy of gabapentin and pregabalin for the treatment of acute mania or acutely depressed BD patients. It should not be used as monotherapy in the short- or long-term period, however, as adjunctive therapy, its effects on the long-term outcomes of BD remain to be elucidated.

## Figures and Tables

**Figure 1 pharmaceuticals-14-00834-f001:**
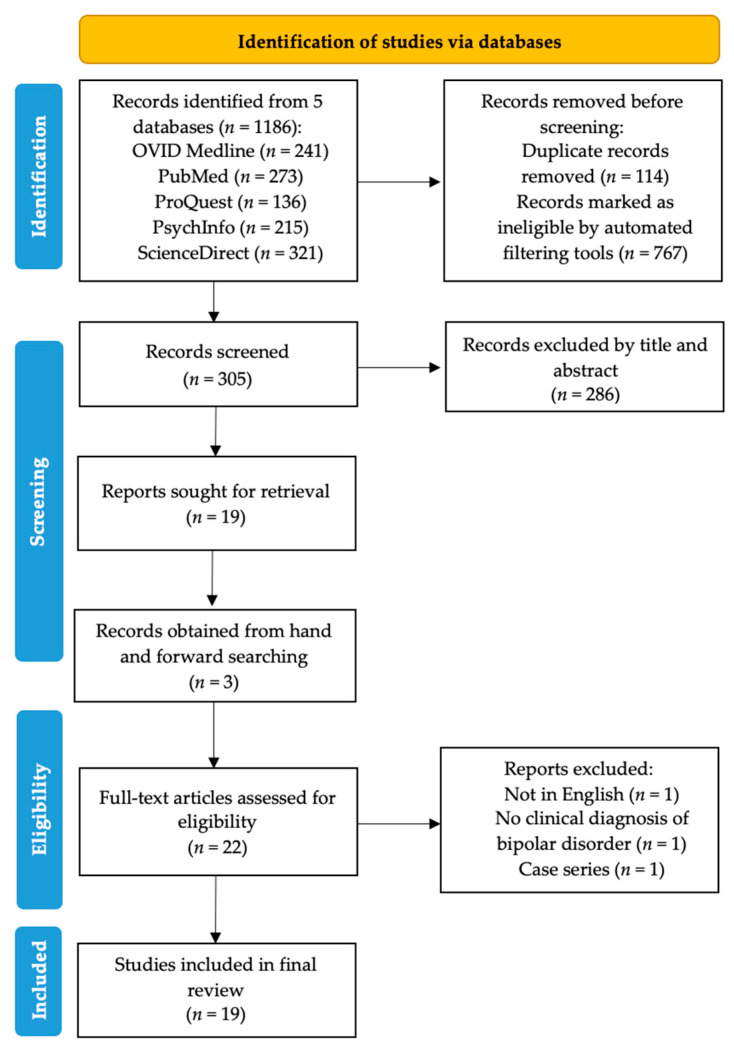
PRISMA flowchart showing the abstraction process.

**Table 1 pharmaceuticals-14-00834-t001:** Studies reviewed (arranged alphabetically by first author’s last name).

Author, Year	Study Design (N)	Study Population	Clinical Assessment or Rating Tool(s)	Intervention(s)	Key Findings
Altshuleret al., 1999 [16]	Open-label clinical trial (*n* = 28 bipolar patients)	Patients had manic (*n* = 18), depressive (*n* = 5) or rapid-cycling (*n* = 5) symptoms that were unresponsive to at least one mood stabiliser. Rapid-cycling patients had treatment initiated when they were euthymic. Patients had consultations with their treating physicians weekly to monthly.	CGI-BP rated every 2 weeks	Gabapentin was given as an adjunctive treatment, added to an existing medicine regimen. Doses ranged from 300 to 3600 mg/day for manic symptoms, 300 to 2400 mg/day for depressive symptoms and 600 to 3000 mg/day for rapid-cycling patients.	- 78% (*n* = 18) of patients treated for manic symptoms had a positive response to a dose range of 600 to 3600 mg/day - Mean times to a recorded positive response was 12.7 ± 7.2 days for hypomania patients, 25 ± 12 days for classic mania and 31.8 ± 20.9 days for mixed mania. - All 5 patients treated for depression had a positive response in 21 ± 13.9 days, while only 1 patient in the rapid-cycling group had a positive response. - 46% (*n* = 12) of patients had reported side effects of sedation (*n* = 5), ataxia (*n* = 2), dizziness (*n* = 1) and headache (*n* = 1)
Astaneh et al., 2012 [17]	Randomized, open-label clinical trial (*n* = 60 bipolar patients)	Patients had a diagnosis of bipolar disorder and were admitted in the acute mania phase.	YMRS rated at the start and the end of therapy	Both groups were treated with lithium for a period of 6 weeks. In the experimental group, gabapentin was given adjunctively (900 mg dose).	- There was significant improvement in the YMRS score of the experimental group as compared to the control group.
Cabras et al., 1999 [18]	Open-label clinical trial (*n* = 25 patients)	Patients were 18 years and older, and had diagnoses of bipolar I disorder (*n* = 16) or schizoaffective disorder (*n* = 9) according to the DSM-IV. Patients also had to fulfill DSM-IV criteria for episodes of mania or hypomania.	CGI-S and BPRS evaluated at baseline and every 2 weeks.	Treatment with gabapentin was given over 16 weeks, with other mood stabilisers tapered off over a period of 4 weeks. Baseline prescriptions of benzodiazepines and neuroleptics were maintained. Gabapentin was administered as an initial dose of 300 mg every night, increased by 300 mg/day every 4 days, titrated to patient response and tolerability (maximal dose of 2400 mg/day).	- 76% of patients (*n* = 19) had a positive response measured by the CGI and BPRS scores. - CGI severity score decreased from 4.0 ± 1.2 at baseline to 2.3 ± 1.1 at week 16. The CGI change score was statistically significant (t = 8.5, df = 21, *p* < 0.0001). - BPRS score decreased from 29.1 ± 7.1 at baseline to 21.3 ± 3.3 at week 16. The BRPS change score was statistically significant (t = 28.2, df = 11, *p* < 0.0001). - The mean dose was 1440 mg/day with over-sedation being the most common side effect, as reported in 44% of patients (*n* = 11).
Carta et al., 2001 [19]	Open-label clinical trial (*n* = 10 patients)	Patients had intellectual disability (ID) with four mild cases, five moderate and one severe. All ten patients had concomitant bipolar disorder (*n* = 6) or schizo-affective disorder (*n* = 4).	Assessment and Information Rating Profile (AIRP) with the psychopathology section derived from the Psychopathology Instrument for Mentally Retarded Adults (PIMRA)	Clinical observations were performed during two separate one-month periods, E0 and E1. Following E0, gabapentin was administered adjunctively with doses ranging from 600 to 900 mg/day. Mean treament duration was four months.	- In five patients with affective disorders, there was a statistically significant decrease in total scores at E1 from E0, with improvements recorded in each scale of psychopathology. - Overall scores were 18.6 ± 1.3 during E0 and 10.2 ± 5.8 during E1 (W = 15, *p* = 0.05, Wilcoxon’s *t*-test). - The statistical significance was only noted for subscales of anxiety, depression and adjustment disorders.
Erfurth et al., 1998 [20]	Open-label trial (*n* = 14 patients)	Patients met the diagnostic criteria for mania according to the ICD-10.	Bech-Rafaelsen Mania Rating Scale (BMRS) assessed on days 0, 3, 7, 14, and 21 by 2 trained psychiatrists not blind to the treatment	Gabapentin was given as monotherapy for 8 patients and adjunctively for 6 patients for up to 21 days. In the adjunctive group, the existing medication had been administered for a minimum of 14 days without a significant improvement in manic symptoms. Gabapentin dose ranging from 1200 to 4800 mg/day.	- Mean BMRS scores decreased significantly from 37.7 to 7.8 in the adjunctive group, and from 27.8 to 9.0 in four patients who completed monotherapy. - Mild sedation was reported as a side-effect in the adjunctive group (*n* = 2) who also had concurrent increases in pre-existing medication; may not be due to gabapentin alone.
Frye et al., 2000 [21]	Double-blind, randomised, crossover trial (*n* = 31 patients)	Patients comprised 18 women and 13 men, with bipolar I (*n* = 11) and bipolar II (*n* = 14) disorder. 23 had rapid-cycling symptoms while 6 were unipolar patients.	CGI-BP, HAM-D, STAI, YMRS and BPRS	Gabapentin was given as an initial dose of 900 mg/day and increased to 1500 mg/day, 2700 mg/day, 3600 mg/day, 4200 mg/day and 4800 mg/day by the end of the first, second, third, fourth and fifth to sixth weeks, respectively. Patients received 3 treatments (gabapentin, lamotrigine and placebo) sequentially over three 6-week phases with an approximate crossover period of one week between phases.	- 26% of patients (8/31) had positive response rates as denoted by the overall CGI rating after gabapentin administration. - The gabapentin response rates for mania and depression were 20% (5/25) and 26% (8/31), respectively. - Common side effects post-gabapentin administration were ataxia (*n* = 3), diarrhoea (*n* = 2), diplopia (*n* = 3), fatigue (*n* = 3) and headache (*n* = 4).
Mauri et al., 2001 [22]	Open-label trial (*n* = 21 patients)	There were 21 outpatients comprising 13 females and 8 males. Patients were diagnosed with bipolar types I and II (according to DSM-IV) and were assessed to be in partial remission. Patients were all intolerant and noncompliant with lithium.	BPRS, HAM-D, HAM-A and Manic Rating Scale (MRS) were assessed at baseline, days 15, 30 and then monthly up to 12 months	Gabapentin was administered at doses ranging from 300 to 2400 mg/day for a period of 1 year. 2 weeks prior to the start of the interventions, all anticonvulsants were ceased, with benzodiazepines used only if necessary.	- Over the one year study period, no significant differences in HAM-D, HAM-A and MRS scores were found. - There was a significant decrease recorded for the mean BPRS scores. - A negative correlation was determined between the dose of gabapentin administered and HAM-A scores (r = 0.16, *p* = 0.035) but no relationship was found with the mean scores of BPRS, HAM-D and MRS. - No relationship between adverse events and gabapentin dose (mg/kg) observed.
McElroy et al., 1997 [23]	Open-label, prospective trial (*n* = 9 patients)	Patients were 18 years and older; diagnosed with either bipolar I (*n* = 7) or II (*n* = 2) according to DSM-IV; did not show adequate response to lithium, valproate or carbamazepine; had symptoms of hypomania, mania or mixed states	Treatment response was evaluated monthly according to this scale: 0 (no response or worsening), 1 (minimal improvement), 2 (moderate improvement) and 3 (marked improvement).	Gabapentin was administered adjunctively as an initial dose of 300 to 900 mg/day, and increased by 300 to 900 mg/day every three to 14 days (titrated to side effects). The maximum dose was 4800 mg/day.	- Seven out of nine patients displayed moderate or marked improvements in symptoms of mania after 1 month of adjunctive gabapentin treatment. - This increased to eight patients after three months of treatment. - Common side effects were sedation (*n* = 7), forgetfulness (*n* = 3) and ataxia (*n* = 2).
Mokhber et al., 2008 [24]	Double-blind, fixed dose, randomised clinical trial (*n* = 59)	Patients comprised 28 women and 3 men; age range between 18 to 60 years; diagnosed with dysphoric mania by DSM-IV; had history of bipolar I disorder with at least one episode of mania prior and a recent episode of mixed mania	Minnesota Multiphasic Personality Inventory 2 (MMPI-2) evaluated at baseline and final visit	Randomisation was performed yielding 3 experimental groups - Group 1 (*n* = 18): gabapentin 900 mg/day - Group 2 (*n* = 20): lamotrigine 100 mg/day - Group 3 (*n* = 13): carbamazepine 600 mg/day	- There was a significant decrease of 50% (*p* < 0.000) in MMPI-2 scores for depression for the group administered with gabapentin. This decrease was higher than the lamotrigine group (33% decrease) and carbamazepine group (13% decrease). - Similarly, a significant decrease of 75% was recorded for the MMPI-2 scores for mania in the gabapentin group. This decrease was higher than the lamotrigine and cabamazepine groups which had reductions of 64% and 59%, respectively.
Obrocea et al., 2002 [25]	Double-blind, three-way, randomized trial (*n* = 45)	35 patients with refractory bipolar affective disorder and 10 patients with refractory unipolar affective disorder were recruited in the clinical study, of which there were 27 women and 18 men.	CGI-BP, HAM-D, clinician and self prospective life chart methodology (LCM), YMRS, STAI and Bunney-Hamburg ratings of depression and mania	Gabapentin was administered at a maximum dose of 4800 mg for 6 weeks with a 1 week interval between two subsequent crossovers to the other agents. Lamotrigine (maximum dose of 500 mg) and placebo (equal number of pills to the other drugs) were also administered for the same duration as that of gabapentin.	- Response rates according to overall CGI-BP were 20/39 (51%), 11/40 (28%), and 8/38 (21%) for patients who were administered lamotrigine, gabapentin, and placebo, respectively. - Younger patients responded better than older patients when gabapentin was administered (r = −0.37; *p* = 0.19). - Patients who had a longer duration of illness responded more poorly than patients who had a shorter duration of illness (r = −0.35; *p* = 0.19). - Patients who were lighter in weight before the trial responded better to gabapentin than those who were initially heavier (r = −0.44; *p* = 0.006). - Patients over the age of 45 years and over 95 kg in weight responded poorly to gabapentin, and some patients displayed worsening symptoms.
Pande et al., 2000 [26]	Double-blind, placebo-controlled trial(*n* = 117)	Study cohort comprised outpatients who were diagnosed with bipolar 1 disorder based on DSM-IV criteria, with manic/hypomanic or mixed symptoms. All included patients had to meet the criteria for a lifetime diagnosis of bipolar I and score of more than or equal to 12 on the YMRS at the first clinic visit.	YMRS, HAM-D, HAM-A, CGI-C, Internal state scale (ISS), and Life Chart for Recurrent Affective Illness	58 patients were administered gabapentin three times a day of a dosage ranging from 600 to 3600 mg/day for 10 weeks, while 59 patients were administered a placebo for the same duration.	- Both treatment groups (gabapentin and placebo) displayed decreased total YMRS scores from baseline to endpoint but this decrease was significantly lower in the gabapentin group (−6) than the placebo group (−9) (*p* < 0.05). - No difference between treatments were observed for the total score on HAM-D - Secondary efficacy measures were similar between the two treatment groups.
Perugi et al., 1999 [27]	Open-label trial (*n* = 21)	Study cohort comprised patients diagnosed with bipolar type I mixed episodes, based on DSM-III-R criteria. Included patients were resistant to therapeutic levels of standard mood stabilizers and the semistructured interview for mood disorder (SIMD) was utilized to ensure that the diagnostic criteria were satisfied.	HAM-D, YMRS and CGI-C	Gabapentin was administered adjunctively starting with an initial dosage of 300 mg/day which was subsequently increased to 2000 mg/day based on the clinical response and occurrence of any significant side effects. The mean (+/− SD) dose of gabapentin at week 8 was 1130 +/− 361.4 mg (range 600 to 2000 mg).	- Out of the 20 patients who completed the 8 weeks of therapy, 10 were regarded as responders: 4 with a CGI score of 1 (marked improvement) and 6 with a CGI score of 2 (moderate improvement); 9 patients were regarded as nonresponders: 7 with a CGI score of 3 (minimum improvement) and 2 with a CGI score of 4 (no change). - 9 of the 10 responders maintained symptomatic remission over a 4 to 12 month period, without adverse effects. - Mean final CGI score for all patients (responders and nonresponders) was 3.7 + 1.1. - Mean HRSD score showed a statistically significant decrease from 18.2 to 10.6 (t = 5.73; *p* < 0.0001).
Perugi et al., 2002 [28]	Open-label trial (*n* = 43)	Study cohort comprised patients diagnosed with bipolar disorder and current major depression (*n* = 14), mixed state (*n* = 24), or a manic episode (*n* = 5), based on the DSM-III-R criteria. The SIMD was utilized to ensure that the diagnostic criteria were met.	HAM-D, YMRS and CGI-C	Gabapentin was given as an add-on therapy with other mood stabilizers for 8 weeks. The initial dosage of gabapentin administered was 300 mg/day which was subsequently increased to 2400 mg/day based on the clinical response and occurrence of any significant side effects. The mean (+/− SD) dose of gabapentin at week 8 was 1272 +/− 465.13 mg (range 600 to 2400 mg).	- 18 patients out of the study cohort were considered responders; 8 had a CGI score of 1 and 10 had a CGI score of 2. - 22 patients out of the study cohort were considered non-responders; 15 had a CGI score of 3, 5 had a CGI score of 4, while 5 had a CGI score of 5. - Mean total HAM-D score showed statistically significant reduction during the 8 weeks of treatment from 16.0 to 8.4 (t = 4.51, *p* > 0.05). - Mean total YMRS score did not show a statistically significant reduction (t = 1.60, *p* > 0.05). - 17 out of the 18 patients deemed as responders maintained symptomatic remission over a period of 4 to 18 months, without side effects.
Schaffer et al., 2013 [29]	Open-label study (*n* = 58)	All patients (46 females and 12 males; mean age 47 years) were outpatients at a private practice and satisfied the DSM-IV diagnostic criteria for bipolar disorder.Patients were administered pregabalin as an add-on therapy if they were deemed to be nonresponders or unsatisfactory partial responders to majority of the standard medications for bipolar disorder.	CGI-BP	The average dose (+/− SD) of pregabalin for acute responders was 72 mg (+/− 69). The average dose (+/− SD) of pregabalin for non-responders was 84 mg (+/−74). The average dose (+/− SD) of pregabalin for patients on maintenance therapy was 90 mg (+/− 67.9).	- 24/58 patients were deemed as acute responders to pregabalin, of which 12 experienced a mood stabilizing effect of either mixed or rapid cycling symptoms; 5 experienced an antimanic effect; 7 experienced an antidepressant effect. - 10 of these 24 patients were still taking pregabalin as an adjunctive therapy for a mean of 45.2 months (range 42–48; SD: 2.35).
Sokolski et al., 1999 [30]	Open-label trial (*n* = 10)	Outpatients (9 males and 1 female, mean age 50.4 years), diagnosed with Bipolar I by SCID. None were psychotic at entry. Previously received therapeutic dosages of mood stabilizers for at least 2 months with partial responses.	HAM-D and Bech Mania Rating Scales (BMRS)	Adjunctive gabapentin 300 mg initially, increased by 600 mg a week until patients reported a full night sleep or could no longer tolerate sedative side effects. Study duration was 10 weeks.	- Adjunctive gabapentin decreased HAM-D and Bech mania rating scores as early as after the first week of study, and the effects were sustained. - Common side effects include somnolence, dizziness and poor coordination, otherwise well-tolerated.
Vieta et al., 2000 [31]	Open-label trial (*n* = 22)	Twenty-two research diagnostic criteria (RDC) bipolar I (*n* = 15) and II (*n* = 7) patients (age >18 years); absence of concomitant serious physical illness; adequate contraceptive control; with presence of at least one episode of the illness in the last six months; presence of residual or subsyndromal features (YMRS > 6 or HAM-D > 12, and CGI-BP > 3); presence of at least one relapse during the treatment with mood stabilizers with serum levels within therapeutic range.	Schedule for Affective Disorders and Schizophrenia (SADS), YMRS and HAM-D scores	Add-on gabapentin were increased by 300 mg/day, titrated to clinical response or tolerance, up to a maximum dose of 2400 mg/day. The mean dose of gabapentin was 1310 mg/day, within a range from 600 to 2400 mg/day. The most common dose prescribed was 1200 mg/day.	- Six patients (27.3%) who did not complete the study dropped out for different reasons: four de to lack of efficacy, one because of intolerance (mild rash) and another because of noncompliance. - 8 patients improved as there was a decrease of at least 2 points in the CGI-BP, in the other eight patients who completed the study, a modest improvement was observed in three of them; four did not show any therapeutic effects. - The comparison of mean scores in CGI-BP showed a significant improvement in the depression subscale that decreased from 4.5 ± 1.2 to 2.9 ± 1.5 points (Wilcoxon Z = −3.1074, *p* < 0.002), taking into account only patients who completed the study. - The improvement in the mania subscale was not significant (3.3 ± 1.1 vs. 2.9 ± 1.0; Wilcoxon Z = −1.5799, *p* = NS). - Most patients showed some improvement in social functioning and irritability. - There were non-significant differences in the efficacy of gabapentin between bipolar I and II patients, and between rapid cyclers and non-rapid cyclers.
Vieta et al., 2006 [32]	Double-blind, placebo-controlled, randomized trial (*n* = 25)	Patients, *n* = 13 in the treatment group (mean age 46.2 years) and *n* = 12 in the placebo group (mean age 47.6 years), diagnosed with bipolar I or II according to DSM-IV criteria and were treated with any standard mood stabilizer in the last year; two bipolar episodes or more during the last year; CGI-BP scores equal or greater than 4; last episode within past 6 months; euthymic; score of 8 or less on the HAM-D and 4 or less on the YMRS.	CGI-BP, HAM-D, HAM-A, PSQI and YMRS	Gabapentin dose was 1200 mg/day and kept that way unless there were emerging symptoms, then it was increased up to 2400 mg/day and if there were adverse events it would be reduced to 900 mg/day.	- 13 subjects (7 from gabapentin group and 6 from placebo group) completed the study. - Reasons for discontinuation in the gabapentin group were due to withdrawal to participate (*n* = 2), lack of efficacy (*n* = 2), adverse events (*n* = 1) and other reasons (*n* = 1). In the placebo group, 3 no longer wanted to participate, 1 had a lack of efficacy and 1 had adverse events and 1 (other reasons). - The change in CGI-BP between the groups were statistically significant (gabapentin:−2.1, Placebo: −0.6, *p* = 0.0046). - No significant differences between groups in YMRS, HAM-D, HAM-A and PSQI scores. - PSQI-6 subscale (use of sleeping medication), the score change at month 12 in the gabapentin group was −1.1 and placebo was −0.6 (*p* = 0.0267).
Wang et al., 2002 [33]	Open-label trial (*n* = 22)	Outpatients (10 females and 12 males, mean age 38.4 years); met DSM-IV Criteria for bipolar I or II disorder by semistructured clinical interview and DSM-IV criteria for major depressive episode with a 28-item HAM-D score >18 at screening	28-item HAM-D, YMRS and CGI-S	Adjunctive therapy of gabapentin to stable doses of mood stabilizers or atypical antipsychotics, initiated at 300 mg at bedtime and increased by 300 mg every four nights until symptom relief or adverse effects were noted. Final GBP dose was clinically determined. Maximum dose 3600 mg per day in divided doses (range 600 mg to 3300 mg).	- There was a decrease in mean HAM-D ratings from 32.5 (7.7) to 16.5 (12.8) (t = 8.11, df = 21, *p* = <0.0001). - Mean CGI-S decreased from 4.4 (0.9) to 3.0 (1.7) (t = 5.2, df = 21, *p* < 0.0001). - YMRS were unchanged. - 12 of 22 patients were responsive to treatment, with mean HAM-D decreasing 78% from 27.9 (6.2) to 6.2 (4.5), *p* < 0.0001. 8 of 22 patients were remitted. In non-responders, HAM-D decreased 24% from 38.0 (5.4) to 28.9 (6.7), *p* = 0.005. - Mild to moderately depressed patients (HAM-D less than 35 at baseline) had a response rate of 77%. Severely depressed patients (HAM-D equal to or greater than 35 at baseline) had a response rate of 22%. Mild to moderately depressed patients had their HAM-D decreased by 16.7 (8.6). Severely depressed patients had their HAM-D decreased by 15.0 (10.7). - Responders had longer bipolar disorder illness duration (23.3 (12.2) vs. 12.9 (9.8)). - Final gabapentin dose was higher in non-responders (2085) than in responders (1425). - Gabapentin was generally well tolerated. Mild sedation was the most common adverse effect in 7 patients.
Young et al., 1999 [34]	Open-label trial (*n* = 37)	Outpatients (12 males and 23 females; mean age 42.2 years), diagnosed with bipolar disorder Type I or II, based on the structured clinical interview for DSM-IV; in both depressed and manic phases. All treated previously with and failed to respond to at least two mood stabilizers.	HAM-D and YMRS	Adjunctive gabapentin to current treatment, dosed 2 to 3 times a day, ranging from 300 mg/day to a maximum of 3600 mg/day and a mean daily dose of 1264 mg (SD: 136).	- Those who were depressed at the start of the study showed a significant decrease in depression symptoms (*p* < 0.001). This improvement was maintained over 6 months in 17 patients. Significant improvement in the global assessment of functioning from baseline to 12 weeks and 24 weeks (55 +/− 1.3 to 67 +/− 2.9 to 67 +/− 3.6). - In maniac patients, there was a reduction in mania symptoms (*p* < 0.001) and maintained over 6 months. The manic group showed nonsignificant improvement in global assessment of functioning. - There was a significant overall reduction in anxiety and mood clusters (*p* < 0.001). - The drug was well tolerated. Side effects include: constipation (*n* = 4), dry mouth (*n* = 6), trouble sleeping (*n* = 7), daytime drowsiness (*n* = 8), anxiety (*n* = 9), blurred vision (*n* = 5), sexual difficulties (*n* = 9).

Abbreviations: BPRS, Brief Psychiatric Rating Scale; CGI-BP, Clinical Global Impression Scale modified for Bipolar illness; CGI-S, Clinical Global Impression of Severity; DSM, Diagnostic and Statistical Manual of Mental Disorders; HAM-A, Hamilton Rating Scale for Anxiety; HAM-D, Hamilton Rating Scale for Depression; ICD, International Statistical Classification of Diseases and Related Health Problems; PSQI, Pittsburgh Sleep Quality Index; SD, standard deviation; SIMD, semistructured interview for mood disorder; STAI, Spielberger State-Trait Anxiety Inventory; Young Mania Rating Scale, YMRS.3.1. Study Designs.

**Table 2 pharmaceuticals-14-00834-t002:** Results of Cochrane Collaboration’s tool for assessing risk of bias.

Study (Author, Year)	Sequence Generation	Allocation Concealment	Blinding	Incomplete Outcome Data	Selective Outcome Reporting	Other Bias
Altshuleret al., 1999 [16]	−	−	−	+	+	?
Astaneh et al., 2012 [17]	+	−	−	?	?	?
Cabras et al., 1999 [18]	−	−	−	+	?	?
Carta et al., 2001 [19]	−	−	−	−	?	?
Erfurth et al., 1998 [20]	−	−	−	?	−	−
Frye et al., 2000 [21]	+	?	+	+	+	?
Mauri et al., 2001 [22]	−	−	−	+	?	?
McElroy et al., 1997 [23]	−	−	−	−	?	−
Mokhber et al., 2008 [24]	?	?	+	?	+	?
Obrocea et al., 2002 [25]	?	?	?	+	+	?
Pande et al., 2000 [26]	?	?	?	+	+	−
Perugi et al., 1999 [27]	−	−	−	?	+	?
Perugi et al., 2002 [28]	−	−	−	?	+	?
Schaffer et al., 2013 [29]	−	−	−	?	?	?
Sokolski et al., 1999 [30]	−	−	−	+	?	+
Vieta et al., 2000 [31]	−	−	−	+	+	+
Vieta et al., 2006 [32]	+	+	+	+	+	?
Wang et al., 2002 [33]	−	−	−	+	?	?
Young et al., 1999 [34]	−	−	−	+	?	?

Key: + low risk of bias; − high risk of bias; ? unclear risk of bias.

## Data Availability

Data sharing not applicable.

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
