# Peer review of "A Systematic Review of the Clinical Use of Gabapentin and Pregabalin in Bipolar Disorder"

_pharmaceuticals, 2021, doi:10.3390/ph14090834_

Round 1
Reviewer 1 Report
This manuscript is an interesting review of the adjunctive treatment of gabapentin and pregabalin in bipolar disorder. The adopted methodology is correct, and the results are clear. The introduction should be focused more on the treatment strategy of bipolar disorder and less on the socioeconomic burden.
Author Response
Thank you for the positive feedback and comments.
In the introduction, we have now added comments on the treatment strategy of bipolar disorder, "Therapeutic measures for BD are unfortunately limited by the incomplete remission of symptoms and frequent relapses. The Systematic Treatment Enhancement Program for Bipolar Disorder (STEP-BD) study reported recurrence rates of more than 50% and recovery rates of lower than 60% [9]. There is a glaring paucity of treatment options for bipolar depression and a clear need for effective acute and maintenance treatments for all individuals with BD, in order to delay illness progression, restore functioning and improve quality of life [10]" and "In modern clinical practice, patients generally are started on first-line BD medications and depending on symptom improvement and tolerability, either continued on the treatment regimen (with appropriate dose titration) or progressed to second-line medications. Alas, a combination of antipsychotics, mood stabilizers and other classes of psychotropic medications is often the choice in this challenging patient population, despite the inconsistent and scant evidence for polypharmacy [11]."
Reviewer 2 Report
This is, in summary, an interesting systematic review aimed to synthesise the available human clinical trials and inform evidence-based pharmacological approaches to BD management. The authors found that a total of 6 randomized, controlled trials (RCTs) and 13 open-label trials involving the use of gabapentin and pregabalin in BD patients were reviewed. Overall, the studies show that gabapentin and its related drug pregabalin do not have significant clinical efficacy as either monotherapy or adjunctive therapy for BD. Finally, gabapentin and pregabalin are presumably ineffective for acute mania based on the findings of RCTs, with only small open-label trials to support its potential adjunctive role.
The authors may fnd as follows my main comments/suggestions.
First, as the authors, throughout the Introduction section, correctly focused on bipolar disorder, they could even briefly mention the link between bipolar disorder and negative clinical outcomes such as suicidal behavior. Importantly, despite the continuous advancement in neuroscience as well as in the knowledge of human behaviors pathophysiology, currently suicide represents a puzzling challenge. Thus, given the importance of this topic (although i understand that the link between bipolar disorder and negative clinical outcomes is not the main topic of the present paper), i suggest to cite within the main text the article published on Int J Mol Sci in 2018.
In addition, the involvement of sensory perception which is implicated in emotional processes, might be further discussed. Importantly, the unique sensory processing patterns of individuals have been reported as crucial factors in determining clinical outcomes. Thus, given the above information, my suggestion is to include within the manuscript, the study published in 2016 on Child Abuse Negl (PMID: 27792883).
Moreover, why the present systematic review has been conducted only on OVID Medline, PubMed, ProQuest, PsychInfo and ScienceDirect and not on other scientific databases such as Cochrane Systematic Reviews or Web of Science is a matter of debate that should be clarified by the authors.
Finally, what is the take-home message of this manuscript? While the authors reported that the long-term outcomes of BD remain to be elucidated, they might, in my opinion, focus in a more detailed manner, on some conclusive remarks of their research. Specifically, what are the main implications of the present systematic review? Here, the authors should provide more detailed explanations based on their expertise. Specifically, do they recommend or not to use gabapentin and pregabalin as monotherapy in the long-term period? Thus, some additional information might be useful for the readers.
Author Response
Point 1: First, as the authors, throughout the Introduction section, correctly focused on bipolar disorder, they could even briefly mention the link between bipolar disorder and negative clinical outcomes such as suicidal behavior. Importantly, despite the continuous advancement in neuroscience as well as in the knowledge of human behaviors pathophysiology, currently suicide represents a puzzling challenge. Thus, given the importance of this topic (although i understand that the link between bipolar disorder and negative clinical outcomes is not the main topic of the present paper), i suggest to cite within the main text the article published on Int J Mol Sci in 2018.
Reply 1: Thank you for the comment. This has now been added in the introduction with the suggested citation, "This predisposes the individual with BD to a significantly higher risk of death by suicide [4], an unfortunate clinical outcome that remains a challenging and pertinent issue [5]."
Point 2: In addition, the involvement of sensory perception which is implicated in emotional processes, might be further discussed. Importantly, the unique sensory processing patterns of individuals have been reported as crucial factors in determining clinical outcomes. Thus, given the above information, my suggestion is to include within the manuscript, the study published in 2016 on Child Abuse Negl (PMID: 27792883).
Reply 2: Thank you for the comment. This has now been added in the introduction section, with the suggested citation, "It has also been suggested that sensory processes unique to individuals are implicated in their corresponding emotional patterns, making BD a very heterogenous condition [6]."
Point 3: Moreover, why the present systematic review has been conducted only on OVID Medline, PubMed, ProQuest, PsychInfo and ScienceDirect and not on other scientific databases such as Cochrane Systematic Reviews or Web of Science is a matter of debate that should be clarified by the authors.
Reply 3: Thank you for the comment. We chose to include only OVID Medline, PubMed, ProQuest, PsychInfo and ScienceDirect as these are the major biomedical databases, and we also had to allocate time for extensive hand searching, which was performed in this review to ensure that our literature search process was thorough and comprehensive.
Point 4: Finally, what is the take-home message of this manuscript? While the authors reported that the long-term outcomes of BD remain to be elucidated, they might, in my opinion, focus in a more detailed manner, on some conclusive remarks of their research. Specifically, what are the main implications of the present systematic review? Here, the authors should provide more detailed explanations based on their expertise. Specifically, do they recommend or not to use gabapentin and pregabalin as monotherapy in the long-term period? Thus, some additional information might be useful for the readers.
Reply 4: Thank you for the comment. We have revised our conclusion to speak more directly to readers, "In conclusion, there is a lack of rigorous evidence to support the clinical efficacy of gabapentin and pregabalin for the treatment of acute mania or acutely depressed BD patients. It should not be used as monotherapy in the short- or long-term period, however, as adjunctive therapy, its effects on the long-term outcomes of BD remain to be elucidated."
Round 2
Reviewer 2 Report
In the revised manuscript, the authors addressed most of the major questions raised by Reviewers improving both the main structure and quality of the present paper. I have no further additional comments.